# Evolution of Microbial Flora Colonizing Burn Wounds during Hospitalization in Uruguay

**DOI:** 10.3390/biomedicines11112900

**Published:** 2023-10-26

**Authors:** Marina Macedo-Viñas, Andrea Lucas

**Affiliations:** 1Centro Nacional de Quemados, Hospital de Clínicas, Piso 13, Avenida Italia s/n, Montevideo 11600, Uruguay; andrealucas1104@gmail.com; 2Molecular Biology and Flow Citometry Laboratory, Pasteur Hospital, Administration of the State Health Services, Larravide 2458, Montevideo 11400, Uruguay

**Keywords:** burns, microbial epidemiology, wound colonization, antibiotic resistance

## Abstract

(1) Background: Infections are a main cause of morbidity and mortality among burn patients. The spectrum of microorganisms depends on the epidemiological context and treatment practices. We aimed to describe the evolution of microbial flora colonizing burn wounds among patients hospitalized during 15 or more days at the National Burn Center in 2015. (2) Methods: Demographic data, length of stay, total body surface area burn, and status at discharge were collected from electronic records and culture results from the laboratory database. (3) Results: Among 98 included patients, 87 were colonized. The mean length of stay was 39 days overall and 16 days in the ICU. *Acinetobacter* spp., *Enterococcus* spp., and *Staphylococcus aureus* predominated. Fifty-six patients harbored multidrug-resistant bacteria and had a significantly greater TBSA. The mean time to colonization was 6 days overall and 14 days for multidrug-resistant bacteria; it was significantly longer for methicillin-resistant *S. aureus* than for methicillin-susceptible *S. aureus*. (4) Conclusions: This is the first report describing the dynamics of microbial colonization of burn wounds in Uruguay. Similarities were found with reports elsewhere, but early colonization with yeasts and the absence of *Streptococcus pyogenes* were unique. Each burn center needs to monitor its microbial ecology to tailor their antimicrobial strategies effectively.

## 1. Introduction

Burn patients have a high susceptibility to infections, mainly due to a disruption of the skin barrier and a dysbiosis of the immune system [1,2]. Infections are a main cause of morbidity and mortality among these patients [3,4,5,6]. Wound infections impair cicatrization, lead to graft loss, and prolong treatment. The microorganisms (MOs) that colonize or infect burn wounds and other anatomical sites originate from the patient’s own flora (endogenous) and from the hospital environment (exogenous) [7,8]. In the last case, multidrug-resistant (MDR) MOs represent a special challenge in the management of patients. 

*Staphylococcus aureus* and *Pseudomonas aeruginosa* are among the most important colonizers and agents of infection [1,9]. *Candida* spp. are the most frequent fungi that originate from the patients’ own flora, while molds are usually from an exogenous origin [10,11].

It is usually described that Gram-positive bacteria are the first MOs that colonize and eventually infect burn wounds, followed by Gram-negative bacteria and, later, by fungi [12]. Nevertheless, the spectrum of MOs depends on the epidemiological context and treatment practices (like topical antimicrobial therapy and early excision) and, thus, varies from one institution to another. For instance, a unique profile was described in a population affected by industrial oil burns [13]. The surveillance of wound colonization allows for the implementation of empiric antimicrobial use policies according to local and regional epidemiology, among other infection prevention and control measures.

The general objective of this study was to analyze the evolution of the microbial flora that colonized the burn wounds of patients admitted to the National Burn Center in Uruguay (CENAQUE) between 1 January and 31 December 2015. The specific objectives were: (a) to describe the epidemiology of microbial flora from wound cultures; (b) to compare the time to colonization (TTC) between different groups of MOs and between MDR and non-MDR bacteria; and (c) to compare the total body surface area burn (TBSA) and the length of stay (LOS) of patients colonized with different groups of MOs.

## 2. Materials and Methods

### 2.1. Patients and Setting

CENAQUE, which is located in the capital city, Montevideo, is the only adult burn center in Uruguay. Patients younger than 18 are occasionally admitted. It is equipped with 6 intensive care unit (ICU) beds, 8 intermediate care beds, and 2 plastic surgery beds. One hundred and fifty five patients were admitted during the year 2015, with a country population of 3,440,000 inhabitants. All patients that were admitted between 1 January 2015 and 31 December 2015 and were hospitalized for 15 or more days were included. The choice of this minimal length of stay (LOS) was due to the fact that since burn wounds are normally sterile in the acute phase, we included patients hospitalized for a length of time that would be enough to get colonized at the center. Of note, the mean LOS at the center was 26 days in 2015. Demographic data, LOS, days in the ICU, TBSA, and status at discharge were collected from the electronic records. 

### 2.2. Microbiology

Culture results from all second- and third-degree burn wound swabs and biopsies were collected through a review of the laboratory records. Routine cultures from burn wounds are performed twice a week to all patients. Additional swabs and/or biopsies are taken when an infection is suspected. Swabs are cultured on tryptic soy agar supplemented with 5% ovine blood and on McConkey lactose agar. Biopsies are quantitatively cultured [14] in the same media plus thioglycolate broth. Agar plates are incubated a 37 °C at room atmosphere and examined at 24 h and at 48 h. In 2015, identification was performed using conventional manual methods [15], and susceptibility testing was performed using the disc diffusion method and the E-test^®^ and interpreted according to CLSI 2015 [16]. For the purposes of this study, a culture was considered “positive” if one or more (up to 3) relevant pathogens were identified, i.e., pathogens known to cause skin and soft tissue infections. “Negative” cultures were those with growth of normal skin flora or polymicrobial flora (more than 3 bacterial or fungal species). In the absence of a primary pathogen, cultures with coagulase-negative *Staphylococcus* spp. or *Corynebacterium* spp. as their main MOs were considered “normal flora”, except if these bacteria grew from biopsy specimens in significant numbers (of 10,000 or more ufc/gram of tissue). Bacteria were classified as MDR or non-MDR according to the criteria proposed by Magiorakos et al. [17]: acquired non-susceptibility to at least one agent in three or more antimicrobial categories. *Enterococcus* spp. were not classified, as not enough antimicrobial classes were tested.

### 2.3. Analysis

Demographic characteristics of patients, LOS, TBSA, outcome, and the evolution of the colonization of burn wounds during hospitalization were described. The rate of positive cultures was calculated over the total performed cultures and the rate of colonization with different MOs was calculated over the total studied population. For analysis purposes, four groups of MOs were considered: Gram-positive cocci (GPC), Gram-negative bacilli (GNB), yeasts, and molds. In some cases, individual genus or species were separately analyzed. Time to colonization (TTC) was defined as the time between hospitalization and the first positive culture. It was described in days for different MOs. The mean TBSA between patients that were colonized and not colonized with MDR MOs was compared using the Student’s *t*-test. Correlation between the TBSA and the number of groups of colonizing MOs, and between the TBSA and the TTC were determined with Spearman coefficient. The mean TTC with non-MDR and with MDR bacteria was compared using the Student’s *t*-test. The number of microorganisms colonizing each patient with respect to their LOS was analyzed with the Chi-square test with R software (version 3.4.4 [18].

### 2.4. Ethical Considerations

This study was conducted following the ethical principles for medical research involving human subjects of the World Medical Association [19]. The protocol was approved by the board of Directors of CENAQUE.

## 3. Results

Ninety-eight patients were hospitalized for at least 15 days during 2015, while only fifty-seven patients stayed for shorter periods, i.e., 63% of all admitted patients were included in this study. 

### 3.1. Demographics and the Total Body Surface Area Burned (TBSA) 

Sixty-five patients were male and thirty-three were female (with a male-to-female ratio of two for a country ratio of 0.9). The mean age was 44 years (range: 16–89 years). The mean TBSA was 16.8% (range: 1–65%).

### 3.2. Outcome

Eighty patients were discharged alive; thirteen were transferred to their medical institution and five died (mortality rate: 5.1%; the overall mortality rate during the study period was 15.3%).

### 3.3. Microbiology

Three thousand and seventy swabs and one hundred and seventy-seven biopsies were performed on the ninety-eight included patients; one thousand one hundred and forty swabs (29.5%) and sixty biopsies (33.9%) were positive for at least one relevant MO; two thousand and twenty-four of the positive samples (swabs or biopsies) grew two or three pathogens, while the remaining nine hundred and seventy-seven were monomicrobial. Eighty-seven patients (88.8%) had at least one positive culture at some point during their hospitalization; seventy-six patients (76.5%) were colonized with Gram-negative bacilli (GNB), sixty-eight (69.4 %) with Gram-positive cocci (GPC), and forty-one (41.8%) with fungi. *Acinetobacter* spp., *Klebsiella* spp., and *P. aeruginosa* were the most frequent GNB observed, while GPC were equally distributed between enterococci and *S. aureus*. *Candida* spp. Were the most frequent fungi observed (Table 1). Eleven patients (11.2%) never had a positive culture. Non-colonized patients had a mean LOS at CENAQUE of 21 days, while colonized patients stayed an average of 41 days (*p* = 0.0039). Twenty-two patients (22%) were colonized with only one group of MOs (GPC, GNB, yeast, or molds), thirty-five (36%) were colonized with two, twenty-seven (28%) were colonized with three, and only three patients (3%) were colonized with the four groups. The most frequent combinations were GPC + GNB and GPC + GNB + yeast. Different morphotypes and susceptibility patterns were found among *P. aeruginosa* isolates. In contrast, most *Acinetobacter* spp. strains were resistant to ampicillin/sulbactam, ceftazidime, piperacillin/tazobactam, imipenem, meropenem, amikacin, ciprofloxacin, and trimethoprim/sulfamethoxazole, were only susceptible to colistin and tigecycline, and had a variable expression of resistance to gentamycin. Most MDR *Enterobacterales* were *Enterobacter* spp. and *Klebsiella* spp., resistant to third-generation cephalosporins (extended spectrum beta-lactamase and/or hyperexpression AmpC mechanisms). Of note, although carbapenemase-producing *Enterobacterales* were first detected in 2010 in Uruguay [20], none of our strains exhibited a phenotype suggestive of this mechanism. There was a direct correlation between the TBSA and the number of groups of colonizing MOs (R = 0.27; *p* < 0.01). Thirty-one of the eighty-seven patients were colonized with non-MDR bacteria (36%), while fifty-six (57%) were also, or exclusively, colonized with MDR MOs. Table 2 shows the number of patients colonized with different genera/species of non-MDR bacteria and their homologous MDR bacteria. There were no significant differences in the mean age and sex between patients colonized with non-MDR MOs and those colonized with MDR MOs (*p* = 0.41 and 0.42, respectively). Patients colonized with MDR MOs had a greater TBSA than those not colonized with MDR MOs (22% vs. 10%; *p* < 0.0001). The mean LOS was longer for patients colonized with MDR MOs than for patients only colonized with non-MDR MOs or not colonized at all (46 vs. 29 days; *p* < 0.0001). The mean LOS in the ICU was 10 days for patients not colonized with MDR MOs and 21 days for patients colonized with MDR MOs (*p* < 0.05).

### 3.4. Time to Colonization

The mean TTC was 6 days for any MO (range from 0 to 37), 9 days for non-MDR bacteria (range from 0 to 54), and 14 days for MDR bacteria (range from 0 to 59) (*p* < 0.05). Fifteen of the eighty-seven colonized patients (17%) were first colonized with a MDR MO. As well as the TBSA, the LOS in the ICU is a surrogate of the severity of the patients; hence, it could be a confounder. 

### 3.5. Time to Colonization with Different Microorganisms

The mean TCC was 17 days for fungi and 18 days for of bacteria. When discriminating via genus or groups, *Staphylococcus*, with *S. aureus* as its only species, and *Acinetobacter* spp. (mostly *A. baumannii complex*) were the first colonizers (with a mean TCC of 16 days), followed by yeast (17 days), molds (18 days), *Enterobacterales* (19 days), *Enterococcus* spp. (20 days), and *P. aeruginosa* (22 days). Figure 1 shows the culture results by week for each patient. MRSA appeared 2 weeks later than MSSA (24 vs. 13 days, respectively; *p* = 0.042), while MDR GNB tended to appear earlier than their non-MDR counterparts (not significant) (Table 3). 

## 4. Discussion

We evaluated part of the CENAQUE population (only those who were hospitalized for at least 15 days) to describe the dynamics of microbial colonization. 

Even though not all studies included comparable populations, the spectrum of MOs and the global mean TTC was quite similar to that described in other studies from different decades [7,12,21,22,23], except for the low rates of *Acinetobacter* spp. in most of them and the absence of *Streptococcus pyogenes* in our study. 

One of the most striking observations of this study was the high prevalence of *Acinetobacter* spp. colonization (54% of included patients). Furthermore, it was found that the TTC with this genus was shorter than the TTC with any other GNB. This was not expected, as *Acinetobacter* is basically a hospital-acquired MO that is likely to occur a long time after admission in debilitated patients. Actually, an outbreak of *A. baumannii* colonization (predominantly at the rectum and skin) started in 2013. It was followed by an endemic situation that finished in 2017. Nevertheless, very few infections were diagnosed. This situation demanded a change in patients care, since most isolates were MDR. It was necessary to introduce antimicrobials, for prophylaxis or for treatment, that were not usually needed (e.g., colistin, rifampin, tigecycline, and doxycycline). These first measures ended with the carrying out of important building works: replacement of the plumbing network, solenoid valves, hoses, faucets, and application of chlorine shock in balneotherapy rooms. Work continued in the intermediate care sector, with the replacement of wooden elements, change in lighting, doors, and windows. Once the works were completed, beds and furniture suitable for hospital cleaning were purchased. The strongest hypotheses for the early colonization with *Acinetobacter* spp. is that it was highly prevalent, it was found in the center’s environment, and it was probably acquired via cross transmission. Although colonization with endogenous flora was more frequent than colonization with exogenous flora (like *Acinetobacter*), it was acquired later during the course of the hospitalization, presumably because it appeared in response to a selection pressure due to the use of extended-spectrum antimicrobials. A study from India showed that *A. baumannii* was also an early colonizer in a setting similar to that described in our center [21]. On the other hand, during an outbreak of *A. baumannii* infections in a burn unit in France, colonization with this MO was detected with a mean TTC of 13 days [24]. 

Overall, more patients were colonized with endogenous flora (*Enterobacterales*, enterococci, *S. aureus*, and *Candida* spp.) than with exogenous flora (non-fermentative GNB and molds). Nevertheless, cross transmission is not negligible. Control of colonization/infection with endogenous flora mostly depends on the rationalization of the use of antimicrobials to avoid the selection of multi-drug-resistant MOs, whereas colonization/infection with exogenous flora depends on multiple other factors, like cleaning of the environment, thorough hand hygiene, and contact precautions to avoid cross transmission. 

As many as 17% of patients were first colonized with a MDR MO. A plausible explanation for this fact is that critical burn patients often receive antibiotics early after their hospitalization to either prevent infections or to empirically treat possible infections suspected via clinical and paraclinical parameters (including high leukocyte counts and fevers) that could be due to other pathophysiological conditions associated with the burns themselves. As described in other studies [7,12,22], *S. aureus*, mainly methicillin-susceptible, were the first colonizers. In contrast, *P. aeruginosa*, which was described as appearing after the sixth day [7,23], was the species that appeared later (beyond the third week), even later than fungi. However, the same TTC that we found for this MO was described in severely burn patients in the Netherlands [25], and a long LOS for this species was also found in a study from Canada [26]. *P. aeruginosa* was only isolated from 23% of patients, that is, half of the patients colonized with *Acinetobacter*. Both *P. aeruginosa* and *Acinetobacter* spp. were described as a frequent cause of infection among burn patients at Colombia’s reference burn center [27]. Recently, *P. aeruginosa* was described as the main GNB rectal colonizer among patients upon admission to a big referral burn center in North Carolina [28]. Although no molecular analysis was performed in our study, given the antibiotic susceptibility patterns described, it is likely that *Acinetobacter* spp. had a clonal origin, while *P. aeruginosa* did not.

More than half of the included patients were colonized with a MDR MO. The overall TTC was significantly longer for the MDR MOs than for the non-MDR MOs, which is similar to what has been described in other studies [26,29], but when comparing individual genus or species, this was only statistically significant for *S. aureus* (MRSA vs. MSSA). In fact, in the case of *Acinetobacter* spp. and *Pseudomonas* spp., MDR strains tended to appear earlier than their non-MDR counterparts, although these differences were not significant. As expected, patients colonized with a MDR MO had a much longer hospital stay than those not colonized with a MDR MO. Hospitalization in the ICU was also associated with a greater risk of colonization with MDR MOs. Patients colonized with specific MDR MOs were subjected to differential care treatments depending on the MO; for example, patients colonized with MDR *Acinetobacter* spp. during the outbreak were assigned specific staff teams. As we did not detect carbapenemase-producing GNB during this study period, no specific measures were needed. Extended-spectrum beta-lactamase (ESBL)-producing *Enterobacterales* are endemic in many centers in Uruguay but were infrequent at CENAQUE in 2015; these patients were not subjected to specific infection control and prevention (ICP) measures, as contact precautions were applied to all patients at the center. It is important to point out that all patients at the center are hospitalized in individual rooms, so contact precautions are easily practicable. More severely ill patients (with a greater TBSA and longer ICU stay) got colonized earlier and with a more diverse flora. The association between a greater TBSA and a longer ICU stay was attributed to the severity. The greater and deeper the burnt area, the easier it is for the MOs to adhere, and in some cases produce biofilms, and persist until the area heals. Measures applied during the early stages of the injury, consistent with debridement of devitalized tissue and hygiene measures regarding wound care, are crucial in the primary management of burn patients to prevent the most frequent complications.

These findings have to be evaluated in the context of the ICP practices in the institution. The ICP team is led by a specialized nurse and integrated via intensive care and plastic surgery physicians and a doctor specialized in microbiology. There is a strict control on hand hygiene and contact precautions that are applied to all patients. At the time of the present study, there was not a general policy on antibiotic use, but a set of rules were established by the ICP team when a particular problem arose. For instance, when the outbreak of *A. baumannii* was detected, a restriction in colistin use was agreed upon, consisting of the authorization by the ICU chief.

The present work has included all patients hospitalized for 15 or more days at the center and followed them throughout the whole hospitalization. It allowed us to monitor the dynamics of colonization. The duration of this study was of one year. Although the number of patients was small compared to other published studies, the Uruguayan population was 3,402,818 in 2015 [30].

As the main limitations, we did not include samples other than wounds, and we were not capable of distinguishing between colonization and infection. Also, these data reflected the situation at CENAQUE in 2015, and it probably changed in the last years. Nevertheless, we believe that this first communication provides an approximation to the knowledge of the epidemiology of the institution. It contributes to the strengthening of infection prevention and control protocols in the event of the appearance of unusual microorganisms or outbreaks of MDR microorganisms. In turn, it constitutes a baseline for continuing surveillance and analyzing long-term trends. Also, we compared our findings with published data from about the same time and older. We expect to add data from 2016 and to publish a long-lasting trend.

## 5. Conclusions

To the best of our knowledge, this is the first report on burn wound colonization in Uruguay and also one of the few published from South America. Although the information dates from eight years, we believe that it is important as a starting point for future surveillance and that it gives valuable information for ICP practices.

Similarities have been found with the epidemiology described both in developed and in developing countries, but particular characteristics were seen in our study. In addition to the high rates of *Acinetobacter* spp. colonization, due to an outbreak during the study period, an early colonization with yeast compared to bacterial pathogens was observed. Also, we did not find *S. pyogenes*. Both characteristics can be explained by the fact that we included patients that were hospitalized for at least two weeks. It is very likely that these patients had broad spectrum antibiotic treatment during that time.

Each burn center must monitor their microbial ecology and establish protocols and guidelines according to it.

## Figures and Tables

**Figure 1 biomedicines-11-02900-f001:**
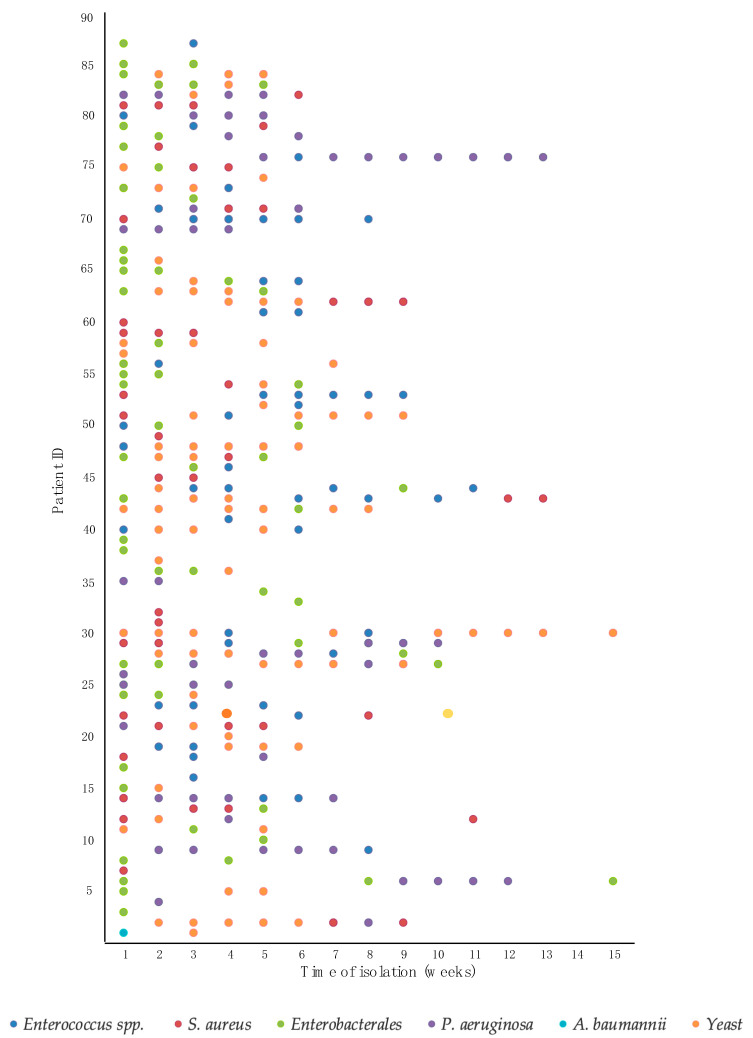
The evolution of colonization with different microorganisms for each patient.

**Table 1 biomedicines-11-02900-t001:** Number of patients colonized with different microorganisms.

Group of Microorganisms	Microorganism		*n*
Gram-negative bacilli (76 patients)	*Enterobacterales* (57 patients)	*Klebsiella* spp.	30
		*Enterobacter* spp.	21
		*Proteus* spp.	18
		*Escherichia coli*	6
		*Morganella morganii*	3
		*Serratia* spp.	2
	Non fermentative (61 patients)	*Acinetobacter* spp.	53
		*Pseudomonas aeruginosa*	23
		Other	4
		*Stenotrophomonas maltophilia*	2
Gram-positive cocci (69 patients)	*Enterococcus* spp.		48
	*Staphylococcus aureus*		45
	*Streptococcus* spp.		2
Fungi (41 patients)	*Candida* spp.		37
	Molds		7

**Table 2 biomedicines-11-02900-t002:** Number of patients colonized with multidrug-resistant and non-multidrug-resistant microorganisms.

Microorganism	Sub-Group of Microorganisms	*n* (Patients)
*Enterobacterales*	Non-MDR	55
	MDR	8
*Staphylococcus aureus*	MSSA	36
	MRSA	13
*Pseudomonas aeruginosa*	Non-MDR	21
	MDR	3
*Acinetobacter* spp.	Non-MDR	7
	MDR	49

MDR: multidrug-resistant; Non-MDR: non-multidrug-resistant; MSSA: methicillin-susceptible *Staphylococcus aureus*; MRSA: methicillin-resistant *Staphylococcus aureus.*

**Table 3 biomedicines-11-02900-t003:** The mean time to colonization with multi-drug- and non-multi-drug-resistant microorganisms.

Microorganism	Sub-Group of Microorganisms	TTC (Days)	*p*-Value
*Enterobacterales*	Non-MDR	18	0.77
	MDR	16	
*Staphylococcus aureus*	MSSA	13	**0.042**
	MRSA	24	
*Pseudomonas aeruginosa*	MDR	12	0.32
	Non-MDR	23	
*Acinetobacter* spp.	MDR	15	0.24
	Non-MDR	20	

## Data Availability

The data presented in this study are available on request from the corresponding author. The data are not publicly available due to ethical restrictions.

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
