# Peer review of "Evolution of Microbial Flora Colonizing Burn Wounds during Hospitalization in Uruguay"

_biomedicines, 2023, doi:10.3390/biomedicines11112900_

Round 1
Reviewer 1 Report
This is an interesting report of microbial flora in burn wounds. Overall the method is appropriate and the results add to the body of knowledge. However it's not clear why the period January 1st to December 31st 2015 was chosen as the review period. That is now 8 years ago so it's uncertain how contemporaneous / relevant the results are. The data certainly provide good background in terms of a baseline, but I do think the study needs replicating to examine whether there is a trend. This is mentioned in the Discussion and I think would be a much more useful piece of work for the reader in terms of examining data from 2015 onwards.
Overall the quality of English is good but there are some aspects of grammar which require attention.
Author Response
Thank you very much for taking the time to review this manuscript. In response to your comment:We carried out the investigation on 2016-2017 but we could not submit the manuscript before, partly because CENAQUE’s authorities changed twice from 2016 to date, and we needed to present the manuscript to new authorities for approval. We are managing the collection of information from recent years but unfortunately it is going to take us a long time since we once again need permits from the center authorities. Certainly, it would be very interesting to continue with this project. Because of the principal author of this paper does not work at CENAQUE anymore, it is difficult to plan forthcoming investigations. Nevertheless, RN Andrea Lucas is still the nurse of the IPC at CENAQUE and surveillance is a main IPC activity. We added to the text a more detailed explanation for this limitation of the study.
Regarding your appreciation about the description of the methods, we did not find specific methodology aspects to improve. If necessary, we will improve them according to new suggestions.
Reviewer 2 Report
1-Could you provide more context on why only patients hospitalized for at least 15 days were included in the evaluation? How might this selection criteria affect the generalizability of the findings?
2-In comparing the spectrum of microorganisms (MOs) and the mean Time to Colonization (TTC) with other studies, were there any specific criteria or characteristics used to select the populations for comparison? How were these populations chosen?
3-The high prevalence of Acinetobacter spp. colonization is noted as a striking observation. What are the potential implications of such a high prevalence in terms of patient care and infection control measures?
4-The observation that Acinetobacter spp. had a shorter TTC compared to other Gram-negative bacteria (GNB) is unexpected. Are there any hypotheses or potential explanations provided for this finding?
5-It is mentioned that an outbreak of A. baumannii colonization occurred in 2013. Were any specific interventions or infection control measures implemented to address this outbreak, and if so, were they effective in controlling further infections?
6-The observation that more patients were colonized with endogenous flora compared to exogenous flora is noted. What are the potential implications of this finding in terms of infection control practices?
7-The study mentions that 17% of patients were first colonized with MDR-MO. Are there any discussions on potential sources or routes of transmission for these multidrug-resistant microorganisms?
8-The association between colonization with MDR-MO and longer hospital stay is mentioned. Were there any specific interventions or measures in place to manage patients colonized with MDR-MO?
9-The study notes that patients with greater TBSA and longer ICU stay were colonized earlier and with more diverse flora. Are there any potential clinical implications or recommendations based on this finding?
10-The limitations of the study are discussed, including the exclusion of samples other than wounds and the inability to distinguish between colonization and infection. Are there any plans to address or mitigate these limitations in future studies?
Author Response
Thank you very much for taking the time to review this manuscript. Your comments helped us to greatly improve the manuscript. Please find the detailed responses below and the corresponding revisions/corrections highlighted in red in the re-submitted file.1-Could you provide more context on why only patients hospitalized for at least 15 days were included in the evaluation? How might this selection criteria affect the generalizability of the findings?
Since burn wounds are normally sterile in the acute phase, we decided to take patients hospitalized for a time enough to get colonized at the centre. We think that our approach limits the possibility of comparison with other studies but gives a better idea about in-hospital dynamics of microbial circulation. Of note, the mean time of hospitalization, considering all patients, was 26 days. One-hundred fifty-five patients were admitted in 2015 and 98 of them (63%) stayed during 15 or more days. Nevertheless, data is not generalizable to patients with shorter length of stay nor to other burn centres. In fact, we concluded that each centre has to maintain its own surveillance.
We added a sentence in the text explaining why we included only patients hospitalized for 15 or more days.
2-In comparing the spectrum of microorganisms (MOs) and the mean Time to Colonization (TTC) with other studies, were there any specific criteria or characteristics used to select the populations for comparison? How were these populations chosen?
There were no specific criteria to select populations for comparison. Not many studies have evaluated the TTC of burn wounds, so we took those available and compared parameters that could be comparable, depending on the study (e.g. TTC with specific microorganisms and global spectrum of MOs).
3-The high prevalence of Acinetobacter spp. colonization is noted as a striking observation. What are the potential implications of such a high prevalence in terms of patient care and infection control measures?
The situation demanded a change in patients care since most isolates were multi-drug-resistant. It was necessary to introduce antimicrobials, for prophylaxis or for treatment, that were not usually needed (colistin, rifampin, tigecycline, doxycycline). On the other hand, aggressive infection control measures were implemented. Detailed measures were added to the text.
4-The observation that Acinetobacter spp. had a shorter TTC compared to other Gram-negative bacteria (GNB) is unexpected. Are there any hypotheses or potential explanations provided for this finding?
The strongest hypothesis is that Acinetobacter was highly prevalent (54% of patients colonized), was found in the centre’s environment and it was probably acquired via cross-transmission. Although endogenous flora was more frequent than exogenous flora (like Acinetobacter), it was acquired later during the course of the hospitalization, presumably because it appeared in response to a pressure selection due to the use of extended spectrum antimicrobials. This explanation was added to the text.
5-It is mentioned that an outbreak of A. baumannii colonization occurred in 2013. Were any specific interventions or infection control measures implemented to address this outbreak, and if so, were they effective in controlling further infections?
Specific interventions were introduced and the outbreak ended in 2017. In 2015 colonization rates were high but infections were rare. In 2022, 77 Acinetobacter spp. were isolated over a total of 1927 isolates (4%), so now-a-days the situation is under control. Detailed interventions were introduced in the text in response to reviewer’s question 3 and are the following:
- Building works: replacement of the plumbing network, solenoid valves, hoses, faucets, and application of chlorine shock in balneotherapy rooms.
- Work continued in the intermediate care sector: replacement of wooden elements, change of lighting, doors and windows.
- Once the works were completed, beds and furniture suitable for hospital cleaning were purchased.
6-The observation that more patients were colonized with endogenous flora compared to exogenous flora is noted. What are the potential implications of this finding in terms of infection control practices?
Control of colonization/infection with endogenous flora depends mostly on the rationalization of the use of antimicrobials to avoid selection of multi-drug-resistant MOs, whereas colonization/infection with exogenous flora depends on multiple other factors like cleaning of the environment, thorough hand hygiene and contact precautions to avoid cross-transmission.
This response is added to the text for better clarification.
7-The study mentions that 17% of patients were first colonized with MDR-MO. Are there any discussions on potential sources or routes of transmission for these multidrug-resistant microorganisms?
A plausible explanation for this fact is that critical burnt patients often receive antibiotics early after hospitalization to prevent infections or to empirically treat possible infections suspected by clinical and paraclinical parameters (leukocytes count, fever) that could be due to other pathophysiological conditions associated with burns themselves. As explained above, potential sources of infections are endogenous flora via selection pressure or cross-transmission of exogenous flora.
This response is added to the text for better clarification.
8-The association between colonization with MDR-MO and longer hospital stay is mentioned. Were there any specific interventions or measures in place to manage patients colonized with MDR-MO?
Patients colonized with specific MDR-MO were subject of differential care depending on the MO. E.g. patients colonized with MDR Acinetobacter during the outbreak were assigned specific staff teams. Because during the study period we did not detect carbapeneamse-producing BGN, no specific measures were needed. ESBL producing Enterobacterales are endemic in many centers in Uruguay and are not subject of specific infection prevention and control measures at CENAQUE. Lastly, it is important to point out that all patients at the centre are hospitalized in individual rooms, so contact precautions are easily practicable.
This response is added to the text for better clarification.
9-The study notes that patients with greater TBSA and longer ICU stay were colonized earlier and with more diverse flora. Are there any potential clinical implications or recommendations based on this finding?
The association between greater TBSA and longer ICU stay is attributed to the severity. The greater and deeper the burnt area, it is easier for MOs to adhere, in some cases produce biofilms, and persist until the area heals. Measures applied in the early stages of the injury, consistent of devitalized tissue debridement and hygiene measures regarding the wound care are crucial in the primary management of burn patients to prevent the most frequent complications.
This response is added to the text for better clarification.
10-The limitations of the study are discussed, including the exclusion of samples other than wounds and the inability to distinguish between colonization and infection. Are there any plans to address or mitigate these limitations in future studies?
Because of the principal authors of this paper does not work at CENAQUE anymore, it is difficult to plan forthcoming investigations. Nevertheless, RN Andrea Lucas is still the nurse of the IPC at CENAQUE and surveillance is a main IPC activity.
Round 2
Reviewer 1 Report
Thank you for the revised manuscript and for your explanation regarding why it has taken a number of years to write the study up and submit for publication. This is now explained in the revised paper.
There are some minor spelling errors which I'm sure will be picked up during the editorial processes.
Reviewer 2 Report
The manuscript is improved properly and can be published in its current state.